# AlGaInAs Multi-Quantum Well Lasers on Silicon-on-Insulator Photonic Integrated Circuits Based on InP-Seed-Bonding and Epitaxial Regrowth

Claire Besancon [1,*], Delphine Néel [1], Dalila Make [1], Joan Manel Ramírez [1], Giancarlo Cerulo [1], Nicolas Vaissiere [1], David Bitauld [1], Frédéric Pommereau [1], Frank Fournel [2], Cécilia Dupré [2], Hussein Mehdi [3], Franck Bassani [3] and Jean Decobert [1]

[1] III-V Lab, A Joint Lab of Nokia Bell Labs, Thales Research and Technology and CEA LETI, 91120 Palaiseau, France; delphine.neel@3-5lab.fr (D.N.); Dalila.make@3-5lab.fr (D.M.); joan.ramirez@3-5lab.fr (J.M.R.); giancarlo.cerulo@3-5lab.fr (G.C.); Nicolas.vaissiere@3-5lab.fr (N.V.); david.bitauld@3-5lab.fr (D.B.); frederic.pommereau@3-5lab.fr (F.P.); jean.decobert@3-5lab.fr (J.D.)
[2] LETI, University Grenoble Alpes, CEA, 38054 Grenoble, France; frank.fournel@cea.fr (F.F.); cecilia.dupre@cea.fr (C.D.)
[3] LETI, CNRS, University Grenoble Alpes, CEA, LTM, 38054 Grenoble, France; hussein.mehdi@cea.fr (H.M.); franck.bassani@cea.fr (F.B.)
* Correspondence: claire.besancon@3-5lab.fr; Tel.: +33-169-415-823

**Featured Application: Compact lasers integrated onto SOI for optical communications.**

**Abstract:** The tremendous demand for low-cost, low-consumption and high-capacity optical transmitters in data centers challenges the current InP-photonics platform. The use of silicon (Si) photonics platform to fabricate photonic integrated circuits (PICs) is a promising approach for low-cost large-scale fabrication considering the CMOS-technology maturity and scalability. However, Si itself cannot provide an efficient emitting light source due to its indirect bandgap. Therefore, the integration of III-V semiconductors on Si wafers allows us to benefit from the III-V emitting properties combined with benefits offered by the Si photonics platform. Direct epitaxy of InP-based materials on 300 mm Si wafers is the most promising approach to reduce the costs. However, the differences between InP and Si in terms of lattice mismatch, thermal coefficients and polarity inducing defects are challenging issues to overcome. III-V/Si hetero-integration platform by wafer-bonding is the most mature integration scheme. However, no additional epitaxial regrowth steps are implemented after the bonding step. Considering the much larger epitaxial toolkit available in the conventional monolithic InP platform, where several epitaxial steps are often implemented, this represents a significant limitation. In this paper, we review an advanced integration scheme of AlGaInAs-based laser sources on Si wafers by bonding a thin InP seed on which further regrowth steps are implemented. A 3 μm-thick AlGaInAs-based MutiQuantum Wells (MQW) laser structure was grown onto on InP-SiO$_2$/Si (InPoSi) wafer and compared to the same structure grown on InP wafer as a reference. The 400 ppm thermal strain on the structure grown on InPoSi, induced by the difference of coefficient of thermal expansion between InP and Si, was assessed at growth temperature. We also showed that this structure demonstrates laser performance similar to the ones obtained for the same structure grown on InP. Therefore, no material degradation was observed in spite of the thermal strain. Then, we developed the Selective Area Growth (SAG) technique to grow multi-wavelength laser sources from a single growth step on InPoSi. A 155 nm-wide spectral range from 1515 nm to 1670 nm was achieved. Furthermore, an AlGaInAs MQW-based laser source was successfully grown on InP-SOI wafers and efficiently coupled to Si-photonic DBR cavities. Altogether, the regrowth on InP-SOI wafers holds great promises to combine the best from the III-V monolithic platform combined with the possibilities offered by the Si photonics circuitry via efficient light-coupling.

**Keywords:** heterogeneous integration; epitaxial growth; direct wafer bonding; semiconductor lasers; silicon photonics

## 1. Introduction

Data traffic exponential growth has pushed forward the demand for high speed, high performance, energy-efficient and cost-effective data transmission [1]. Due to the large number of transmitters required, the current InP-monolithic integration platform used to fabricate photonic integrated circuits (PICs) is challenged [2]. The silicon (Si) photonics platform holds a great promise to ensure PICs fabrication scalability since it allows benefitting from the mature complementary metal oxide semiconductor (CMOS) industry [3]. However, it is highly challenging to fabricate a laser source only from Si due to the intrinsic nature of its indirect bandgap. Alternatives based on strained Ge, Ge-Sn materials grown on Si wafers have permitted band-gap engineering and even laser demonstration [4], but performance is still far behind the one that can be obtained in the III-V conventional approach.

III-V integration on Si is facing a booming development phase since the last 30 years due to the perspectives offered by combining III-V's emitting properties with Si-photonics. Hybrid integration of finished III-V laser die on Si photonic chip using flip-chip technique is the current commercialized solution [5]. It has the advantage of independent optimization of III-V and Si parts. However, both chips do not benefit from the same fabrication line. Assembly costs are also not optimized since it requires butt-coupling. Therefore, the full potential of Si photonics is not exploited. III-V-on-Si heterogeneous integration platform consists in bonding a III-V gain region, previously grown on InP wafer, on a silicon-on-insulator (SOI) wafer [6,7]. This approach benefits from low-loss evanescent light coupling therefore reducing packaging costs. It can also take full advantage of large-scale Si fabrication process since the latter is done on large Si wafers [8]. Another attractive strategy consists in the direct growth of III-V gain materials on large Si wafer. In this approach, epitaxy and fabrication can be done using Si large-scale fabrication line. However, it is the most exposed to physical incompatibilities between III-V and Si materials in terms of lattice mismatch, polarity and coefficients of thermal expansion (CTE) differences inducing dislocations, AntiPhase Domaines (APDs) and cracks [9,10]. In order to overcome these limitations, two main strategies have been developed. The first one consists in a planar growth using intermediate buffer layers in order to minimize the dislocations propagation through the active region [11]. In 2019, a successful laser demonstration based on a GaInAsP-MultiQuantum Well (MQW) active structure grown on Si was achieved under continuous-wave (CW) regime [12]. The latter involved a 3.9 µm-thick buffer layer enabling the reduction of dislocation density to $10^8$ cm$^{-2}$. The use of quantum dots (QDs) active region is favorable since it is much less sensitive to dislocations as compared to MQWs structures [13,14]. Impressive laser demonstrations have been achieved with low threshold and high maximum power operation [15]. However, the use of a thick defective buffer layer underneath the active region may compromise efficient light coupling through Si waveguide. The second strategy consists in a selective regrowth of small-volume III-V materials on patterned SOI wafers to block dislocation propagation [16–18]. In this approach, there is no need for a thick buffer layer. Although high quality material has been achieved, the growth of complex MQW-based active region seems challenging. For now, no electrically pumped laser has been demonstrated. What is more, light coupling quality might be impacted if light has to go through a highly defective zone. Another emerging approach consists in transferring III-V finished components onto Si chips using micro transfer printing (µTP). Proof-of-concept PICs demonstrations have successfully been achieved [19]. However, µTP not only requires cost-effective bonding alignments but is also limited to the III-V fabrication process line.

More recently, an advanced heterogeneous scheme based on wafer-seed-bonding and epitaxial regrowth has emerged. The latter allows us to combine the advantages of the monolithic III-V/Si approach without the two major dislocation origins (APDs, lattice mismatch) thanks to the bonding process. Moreover, the possibility to use a thin bonding layer enables efficient light-coupling and therefore complex III-V/Si PICs. This approach was first pioneered by NTT group, whose approach is based on a MQW active region

integrated on Si wafer by bonding followed by lateral InP regrowth to form an horizontal p-i-n junction [20]. Regrowth of MQW-based active region was also successfully done to achieve band-gap tuning by Selective Area Growth technique (SAG) [21]. Altogether, this approach enabled impressive device performance [22]. However, it has been reported that crystal degradation is observed for a 350 nm-thick structure regrown on bonded template due to the thermal strain induced by the difference of CTE between III-V and Si materials [20]. Therefore, it requires an unconventional process based on an extra growth step to form the thin lateral p-i-n junction as compared to the traditional InP monolithic approach usually based on a single growth step to form the μm-thick vertical p-i-n junction. Altogether, this approach cannot be easily applied for most commercial epitaxial providers and consequently its scalability is jeopardized. In the meantime, other groups have developed an approach based on InP-seed-bonding and regrowth of μm-thick vertical p-i-n junctions [23–25]. The ambition is to create a generic integration scheme combining the best offered by the III-V and the Si photonics platforms. The regrowth capability gives access to the large epitaxial toolkit available in the conventional InP monolithic platform, where several epitaxial steps are often implemented [26–28]. For example, buried lasers, as compared to ridge waveguide lasers, are highly beneficial to reduce power consumption and enhance thermal stability [29,30]. In this context, HP group have reported pulsed and CW laser demonstrations up to 20 °C based on the regrowth of a 2.5 μm-thick GaInAsP-MQW laser coupled to a Si-waveguide [31]. Their approach involves InP-Si bonding using Vertical Outgassing Channels (VOCs) buried under the III-V stack to absorb the generated gas produced at the bonding interface [10]. Then, these VOCs have to be opened up by etching the III-V stack in order to be compatible with MOVPE annealing temperature [32], which induces alignment constraints and more complexity to the fabrication process.

In this context, we developed a novel integration scheme based on InP-seed-bonding and regrowth using an oxide layer at the bonding interface thick enough to act as a hydrogen reservoir [33]. Therefore, our platform involves μm-regrowth on a planar InP-seed layer bonded on a SOI wafer. In this paper, we first introduce our fabrication process to integrate AlGaInAs-based lasers on SOI wafer by InP-seed-bonding and regrowth (Section 2). Then, Section 3 provides the development of a 3 μm-thick AlGaInAs-based laser structure grown on InP-SiO$_2$/Si (InPoSi) wafer compared to the same structure grown on InP wafer as a reference. After assessing the thermal strain induced in the structure grown on InPoSi by in situ curvature measurements, morphological characterizations as well as broad-area lasers static performance are presented. Section 4 presents the implementation of the SAG technique in our platform. A 155 nm-wide AlGaInAs-MQW based laser array covering the C+L band fabricated from a single-step SAG process on InPoSi is described. Finally, Section 5 describes the first demonstration of an AlGaInAs regrown laser efficiently coupled to Si-photonics cavities.

## 2. Fabrication Process

The process flow to fabricate a III-V vertical laser source on SOI by means of InP-seed-bonding and regrowth is described in Figure 1. First, a 100 nm-thick InP layer followed by a 200 nm-thick InGaAs sacrificial layer are grown on a 2 inch InP substrate. In the meantime, a 200 mm SOI wafer is fabricated at CEA-LETI's 200 mm line using a standard process with a 500 nm-thick Si top layer in the initial SOI wafer, on which passive rib waveguides are formed by etching the silicon layer [34]. Additional etching steps can be applied to form both 300 nm thick strip and rib waveguides, DBR gratings and fiber grating couplers (FCG). Finally, a silica layer is deposited and a chemical-mechanical polishing is applied to planarize the SOI wafer.

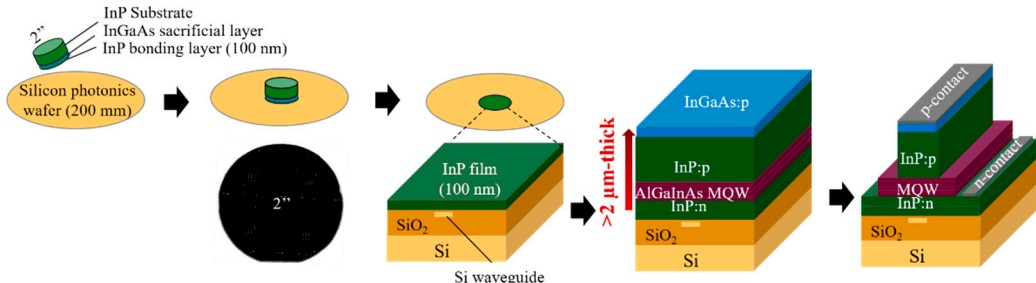

**Figure 1.** Fabrication process for a vertical laser structure on InP-on-SOI platform with an insert acoustic image of the 2 inch InP/InGaAs/InP(substrate) bonded on a 200 mm SOI wafer.

The III-V and SOI surfaces are treated under ozone and nitrogen ($N_2$) plasma treatments. Then, the III-V wafer is bonded on the SOI at room temperature at ambient air in a clean room environment, using hydrophilic Van der Waals forces (hydrogen bonds). The bonded stack is heated up under $N_2$ at 300 °C for 2 h in order to increase the covalent bond ratio. A typical acoustic image of the bonded area (2 in) is shown in Figure 1. The wafer is resized to 3 inch and then the InP substrate material as well as the InGaAs sacrificial layer are selectively removed by chemical etching. Then, a vertical laser structure composed of an AlGaInAs-based MQW active region surrounded by two n- and p-doped InP cladding layers was grown by MOVPE. Finally, shallow-ridge structures are fabricated using the standard technological process developed in our platform [35].

## 3. Development of a 3 μm-Thick Laser Structure Grown on InP-SiO₂/Si

In this platform, we aim at transposing the existing know-how of the InP monolithic platform into the heterogeneous III-V/Si platform, based on the regrowth of μm-thick vertical III-V structures. Therefore, in this section, we present our approach to develop a 3 μm thick AlGaInAs-based MQW laser structure regrown on InP bonded template as compared to the same structure grown on InP wafer as a reference. In order to evaluate III-V laser performance, we have chosen a III-V stack which is not affected by the influence of the passive functions onto the optical mode. Therefore, the regrowth was performed on InPoSi test vehicles in order to mimic the thermal effects of the substrate without any influence of passive functions. In addition, the InP buffer thickness was chosen relatively high to confine to the optical confinement the mode above the InP buffer layer so that the SiO₂/Si substrate contribution is negligible. This brings a relevant comparison between the two lasers.

Thermal strain induced at growth temperature on the structure grown by the InPoSi wafer on the III-V growing layers was assessed from in situ curvature measurement. In situ reflectance signals at different wavelengths were also conducted during the same timeline. Then, ex situ characterizations were carried out to confirm the material quality. Based on these structures, broad-area lasers were fabricated and tested.

### 3.1. Thermal Strain Evaluation at Growth Temperature

The growth on an InP bonded seed enables us to prevent dislocations formation, contrary to direct growth onto Si where high lattice mismatch and APD lead to defective crystal quality. However, this approach still has to face the difference of CTE between III-V and Si. During growth, the lattice of the InP bonded layer expands and follows the CTE of Si bulk material. Therefore, the lower CTE of Si as compared to InP one induces a compressive thermal strain on the III-V grown materials. A 3 μm thick structure composed of a 500 nm-thick n-doped InP layer followed by an AlGaInAs MQW active region, a 2 μm-thick p-doped InP cladding and 200 nm-thick p-doped InGaAs contact layer were grown simultaneously on InPoSi and on InP substrates. The optical confinement is 8% in the MQW for both structures.

In this section, we present the thermal strain evaluation at growth temperature assessed by wafer curvature measurements. The MOVPE system is a CCS from Aixtron with a 6 × 2 inch capacity, equipped with Laytec EpiCurve TT sensor measuring the reflectance signals at 405, 632 and 950 nm and the wafer bowing, which is highly sensitive to the strain induced in the growing layers. The curvature $\frac{1}{R}$ measured simultaneously on InP and on InPoSi is shown in Figure 2. Since the MQW heterostructure was calibrated to be strain-compensated and lattice matched to InP at growth temperature, the curvature signal is flat on InP wafer during growth, contrary to the structure grown on InPoSi. The curvature slope during growth is negative, which is coherent with a compressive strain. The relative curvature value is $-15.8$ km$^{-1}$. Using Stoney Equation (1), the stress $\sigma_{film}$ was assessed. For simplification, the whole III-V stack, including the bonding layer, was associated to InP material ($h_{film}$ = 3044 nm), on Si substrate ($h_{substrate}$ = 525 μm), with a negligible influence of the silica layer on the total strain induced on the III-V stack. Using elastic deformation Equation (2), the evaluated strain $\varepsilon_{film}$ assessed is 400 ppm at growth temperature. This value is coherent with the one previously obtained from the relative curvature value measured on a simplified structure grown on InPoSi [36].

$$\Delta \frac{1}{R} = \frac{\sigma_{film} * 6 * h_{film} * (1 - \nu_{sub})}{h_{sub}^2 * E_{sub}} \tag{1}$$

$$\varepsilon_{film} = \frac{\sigma_{film} * (1 - \nu_{film})}{E_{film}} \tag{2}$$

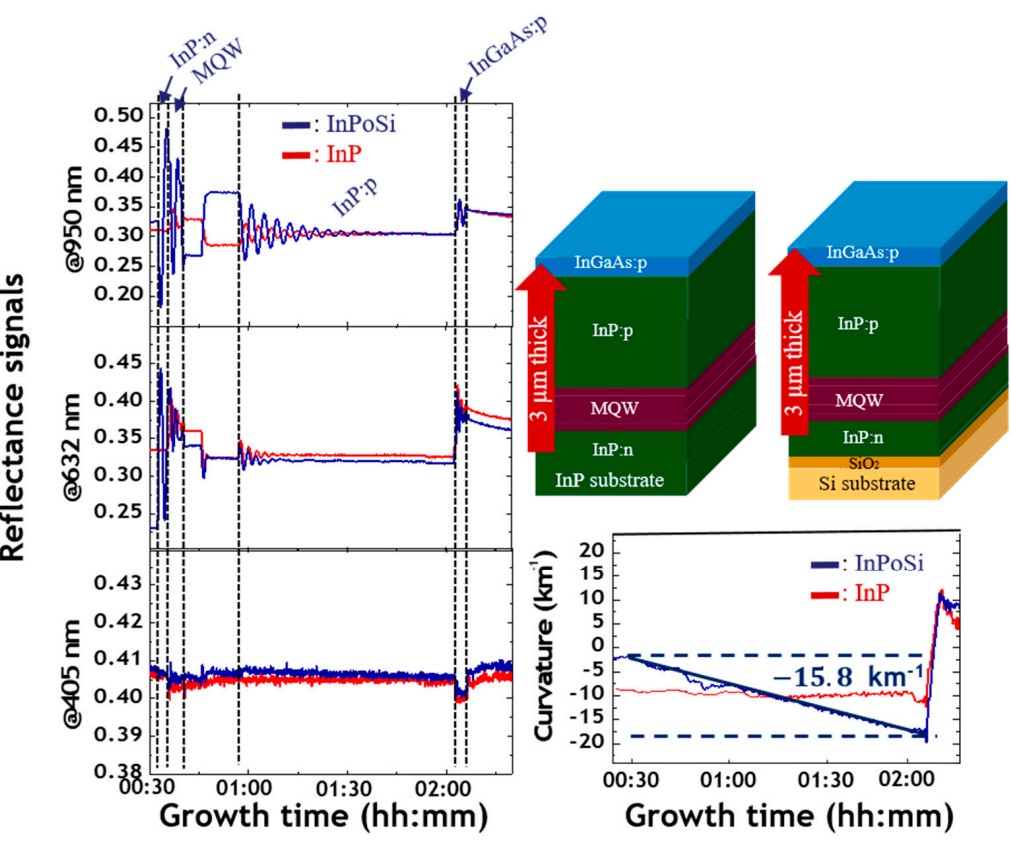

**Figure 2.** In situ measurements during growth: reflectance signals measured at 950, 632 and 405 nm and curvature measurement.

### 3.2. Reflectance Measurements at 405, 632 and 950 nm

The reflectance signal intensity is directly affected by the surface quality of the growing material. Therefore, if the material quality decreases during growth, the signal intensity will decrease instantly. Reflectance measurements during growth are presented on Figure 2.

The highest wavelength reveals the in-depth material quality as compared to the lowest wavelength which is much more sensitive to surface roughness. The mean value of the signal measured at 950 nm is modulated by Fabry-Pérot (FP) oscillations that originate at the interferences between layers/substrate. High amplitude oscillations are particularly noticeable in the InP buffer layer grown on InPoSi. These are induced by the high index contrast in InP/SiO$_2$/Si stack. The signals at 950, 632 and 405 nm are stable, denoting an excellent material quality of the structures grown on both InP and InPoSi substrates. Altogether, reflectance measurements do not reveal any degradation of the regrown materials on InPoSi in spite of the thermal strain.

### 3.3. Material Quality Evaluation

In order to confirm the observations made from in situ measurements, extensive ex situ characterizations were done on both structures by Atomic Force Microscopy (AFM), Scanning Transmission Electron Microscopy (STEM), High-Resolution X-ray Diffraction (HRXRD) and Photoluminescence (PL) measurements. The roughness measured by AFM from a 5 × 5 μm$^2$ surface is of 0.3 nm for the structure grown on InPoSi (Figure 3a). The latter is slightly higher than the one measured for the structure grown on InP (0.2 nm). The visible atomic steps are a signature of high crystal quality. The cross-sectional STEM image demonstrates planar regrowth of the full laser stack on InPoSi (Figure 3b). The PL measurements were done using laser excitation at 1064 nm and an InGaAs detector (Figure 4a). The PL signal peak at 1515 nm measured on InPoSi is 5 times more intense than the one measured for the structure grown on InP. This observation was already made before [20,23,36]. The difference of intensity can be explained by the enhancement of increased excitation and collection induced by the resonant cavity InP/SiO$_2$/Si. The HRXRD profiles are presented in Figure 4b. The satellite peaks are sharp and intense which is a signature of high crystal quality. Altogether, these characterizations demonstrate the excellent material quality of the 3 μm-thick laser structure grown on InPoSi as compared to the one grown on InP as a reference.

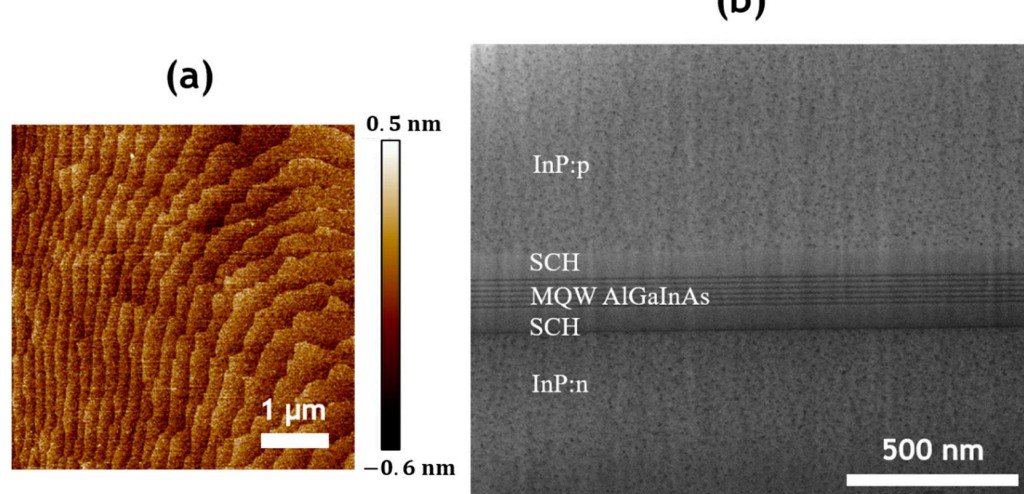

**Figure 3.** Characterizations done on the laser structure grown on InPoSi: (**a**) AFM image done of a 5 × 5 μm$^2$ surface; (**b**) cross-section STEM image.

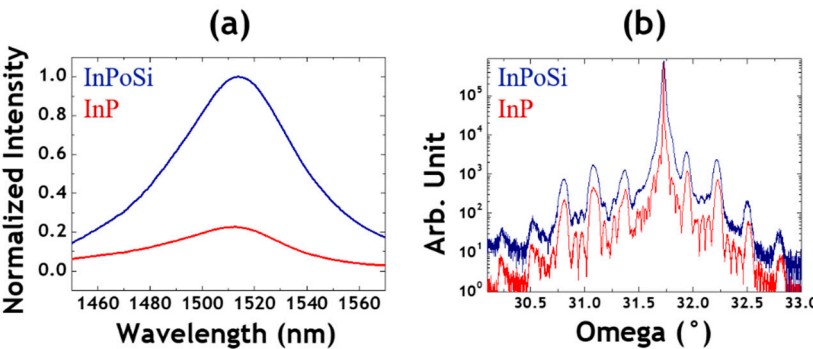

**Figure 4.** Characterizations done on the laser structures grown on InPoSi (blue) and on InP (red): (**a**) normalized intensity of the photoluminescence signals; (**b**) HRXRD profiles.

### 3.4. Broad-Area Laser Fabrication

Based on the structures grown on InPoSi and on InP, broad-area lasers were fabricated. These test vehicles provide a link between the laser performance and material quality thanks to the simplicity of their fabrication process. A 50 μm-wide Pt/Au metallic contact were deposited on top of the p-doped InGaAs layers. Then, wet selective etchings were done to remove the p-doped InGaAs layer, p-doped InP cladding layer and the AlGaInAs-based active region. Finally, a 30 μm-wide Ti/Pt/Au metallic stack was deposited on top of the n-dope InP contact layer. Both components were then cleaved to form a 500 μm-long FP cavity and soldered on thermally conductive submounts. An observation by Scanning Electron Microscope (SEM) of a finalized broad-area laser grown on InPoSi is shown in Figure 5.

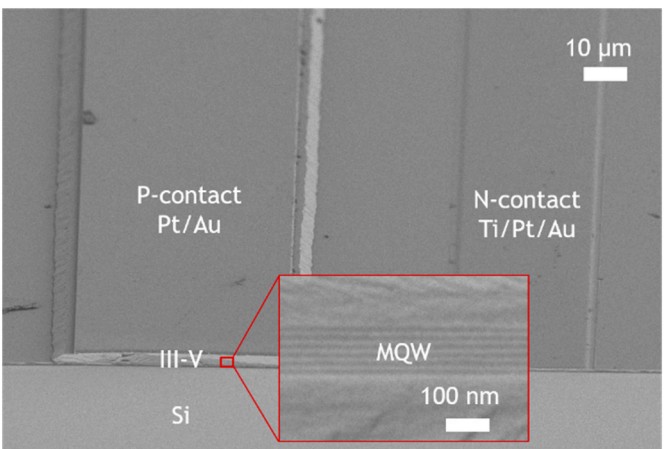

**Figure 5.** Cleaved bar of a broad-area laser on InPoSi observed by SEM with an insert observation in the MQW region.

### 3.5. Broad-Area Laser Performance Comparison

During the fabrication process, the contact resistivity values of the n-doped InP and the p-doped InGaAs contact layers were measured using a transmission line model. The n-doped contact resistivities were $1.5 \times 10^{-6}$ $\Omega.cm^2$ for both structures. The p-doped contact resistivity values were respectively $8.6 \times 10^{-6}$ and $1.6 \times 10^{-6}$ $\Omega.cm^2$ for the structures grown on InPoSi and on InP. This difference was explained by the higher Zn doping level in the InGaAs contact layer for the structure grown on InP as compared to the one grown on InPoSi. The latter is due to growth temperature differences measured between the two substrates by pyrometry during growth, induced by their difference of thermal conductivity. Since Si material has a higher thermal conductivity as compared to InP one, for a given temperature setpoint, growth temperature is 7 °C higher on InPoSi than on InP.

The light-current densities measurements are presented in Figure 6a. These measurements were done by means of a broad detector composed of a Germanium photodiode with an integrating sphere and a source Keithley 2520 used in pulse regime (duty cycle 0.5%) in order to limit thermal effects typically observed in broad area lasers. Performance were assessed at 20 °C using a thermo-electric cooler system. Threshold current densities are respectively of 0.4 kA/cm$^2$ and 0.7 kA/cm$^2$ for the laser grown on InPoSi and on InP. The difference observed, even if not fully inderstood, can be attributed to the different quality of the cleaved facets from both substrates. The latter could have an impact on the losses associated to the reflection in the facets. It could also be attributed to different p-doping profiles due to the difference of growth temperature. Higher Zn doping level on the p-cladding layer of the structure grown on InP could enhance internal losses. The output powers are in the same range of values. The slope efficiency is respectively of 0.092 W/A and 0.095 W/A for the laser on InPoSi and for the one on InP. The temperature dependent behavior of the laser was tested up to 50 °C (Figure 6b). The characteristic temperature $T_0$ is of 53 °C for both lasers. Altogether, the performance similarities between the two lasers confirm that the material quality is similar for the structure grown on InPoSi as compared to the reference.

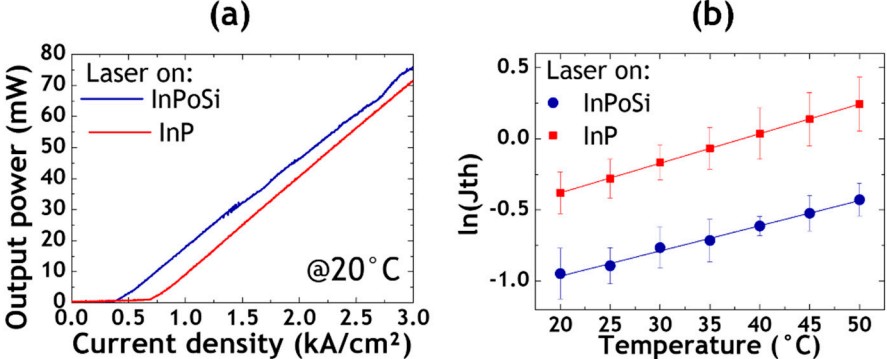

**Figure 6.** Light-current characteristics comparison between the laser grown on InPoSi (blue) and for the one grown on InP (red): (**a**) at 20 °C; (**b**) evolution of the threshold current density up to 50 °C.

## 4. Selective Area Growth Applied on InPoSi Wafers

The main objective of our advanced integration platform based on InP seed-bonding and regrowth is to extend the mature building-blocks from the InP platform in the III-V/Si heterogeneous platform. Among them, the Selective Area Growth (SAG) technique by MOVPE requires an epitaxial step that must be done after the bonding step because of the bonding process specifications (low roughness, low bow). By the precise design of mask geometry, generally two dielectric mask stripes on both side of the SAG region, different active regions, in terms of thicknesses and compositions, can be grown on the same wafer from a single epitaxy step [3–5]. Due to the significant reduction of fabrication complexity, the introduction of SAG into the heterogeneous silicon photonics platform would enable high flexibility in complex PICs.

SAG has been widely used for wavelength detuning both for "longitudinal" and "transversal" integrations of components. In the "longitudinal" integration, along the optical waveguide direction, active and passive components such as electro-absorption modulators (EAM), lasers, semiconductor optical amplifiers (SOA) and spot size converters have been successfully integrated [37–39]. SAG has also demonstrated its high potential for "transversal" integration, with large wavelength shift of numerous juxtaposed devices such as lasers in an array [27,40]. In this context, Coarse Wavelength Division Multiplexing (CWDM) systems are key components for which the SAG technique is particularly adapted. The large inter-channel wavelength shift of 20 nm needed between the laser in the array require a 140 nm-wide PL extension for a 8-channel CWDM system. With respect to a more traditional approach based on discrete DFB lasers fabricated from different epitaxial

growths further integrated on a SOI wafer by means of die-bonding [41], SAG technique allows us to obtain the entire set of emission wavelength from a single growth step.

In this section, we present a 5-channel laser array achieved by SAG on Si wafer. The aim of this study is to evaluate the performance of SAG based active regions without any influence of the passive functions underneath the SOI. Therefore, AlGaInAs-based MQW laser structures were selectively regrown on InPoSi. First, selectively grown AlGaInAs MQW-based structures were characterized by HRXRD and PL measurements. Based on these structures, the 5-channel laser array was fabricated and performance was assessed under CW operation.

### 4.1. Selective Area Growth Fabrication Process on Si Wafers

A SAG process was specifically developed on InP-SOI and on InPoSi wafers. The latter is presented in Figure 7. Instead of depositing a silica layer to achieve growth selectivity, as for the traditional SAG process, the silica from the SOI wafer was locally digged out by the etching of the InP layer in order to open the variable-sized dielectric surfaces for the SAG process. By adjusting the width of the opening ($W_{m,i} \ldots W_{m,n}$), the thickness of the MQW structure, and, therefore its photoluminescence emission, can be tuned ($\lambda_i \ldots \lambda_n$). This specific SAG process enables the growth of the µm-thick III-V structures. The region far from the masks' influence is defined as the "field area", where the thickness and composition of the grown materials are nominal.

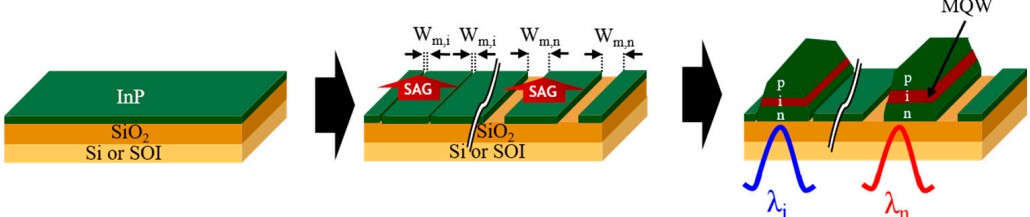

**Figure 7.** Description of the specific SAG process developed on InP-SOI and InPoSi wafers.

### 4.2. Demonstration of Selectively Grown MQW Structures

Selectively grown MQW active regions on InPoSi were thoroughly characterized. In order to cover a broad PL range, mask's width $W_m$ of ranging from 5 µm to 30 µm by step of 5 µm were used, while the distance between the masks was set to be 30 µm. Therefore, AlGaInAs-MQW based active regions surrounded by two undoped InP layers were selectively grown on InPoSi and analyzed by HRXRD, PL measurements and TEM. The SAG regions are 30 µm-large, which require specific means of characterizations to investigate their crystal quality.

First, PL measurements were done using a Horiba-Labram set-up equipped with a microscope to focus the laser excitation spot on the surface of each SAG region (Figure 8a). A 160 nm-wide wavelength extension ranging 1490 to 1650 nm was achieved. The full width at half maximum (FWHM) of the PL signals is almost constant over the entire spectral range. The latter is a clear signature of high-crystal quality of the MQW for the different SAG regions. A significantly lower signal-to-noise ratio is measured for the PL peaks at 1650 nm because of the limited responsivity of the InGaAs detector of our PL setup for wavelengths above 1600 nm.

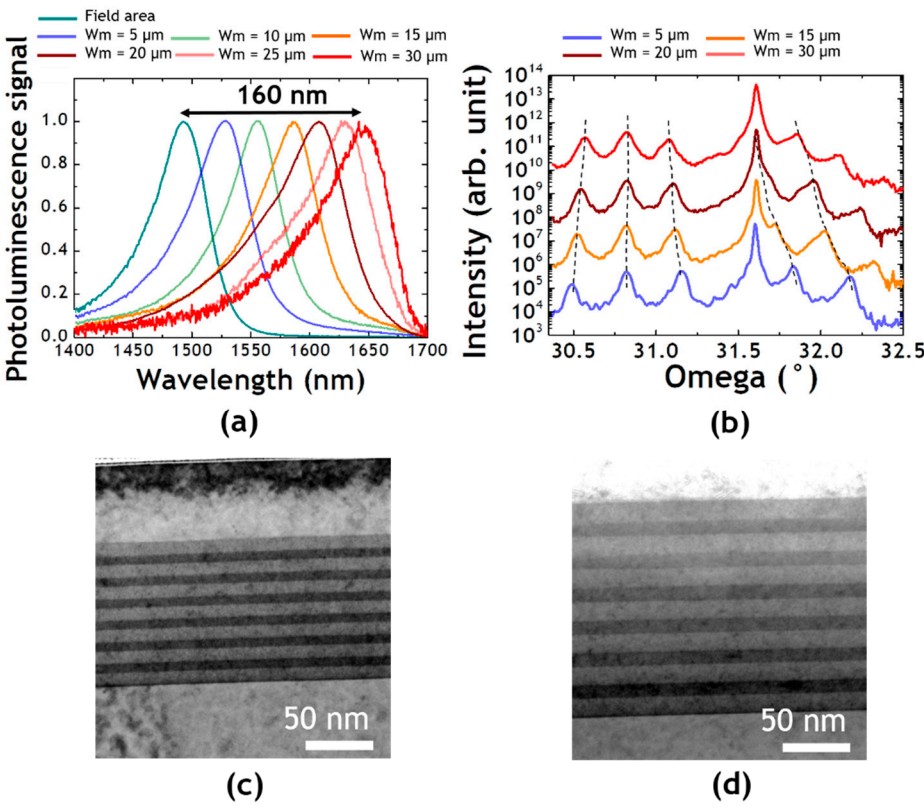

**Figure 8.** Morphologic characterizations of the selectively grown AlGaInAs-based MQW structures on InPoSi: (**a**) PL measurements for SAG masks' widths ranging from 5 µm to 30 µm; (**b**) HRXRD profiles obtained with a micro-focus X-ray source; Cross-section TEM image obtained in SAG regions obtained with (**c**) $W_m$ = 5 µm; (**d**) $W_m$ = 30 µm.

XRD is particularly relevant to characterize the thickness enhancement induced in the selectively regrown MQW structures. Therefore, HRXRD was performed using a D8 Discover diffractometer from Bruker equipped with a microfocus X-ray source (IµS) delivering a sub millimeter spot size and down to 50 µm. The HRXRD diffractograms of the selectively grown MQW structures obtained for $W_m$ = 5, 15, 20 and 30 µm are presented in Figure 8b. The lowering distances between the satellite peaks account for the enhancement of the thickness of the quantum wells and barriers with the SAG mask width. The quantum well and barrier period ranges from 16.4 nm for $W_m$ = 5 µm to 20.4 nm for $W_m$ = 30 µm, with a reference period value determined outside SAG regions of 14.2 nm. One can notice the enlargement of the satellite peaks. The latter is due the contribution of lateral variations in the analyzed region between the masks [42]. However, for a given SAG region, the central zone thickness/composition variation is negligible. Therefore, we can consider that if the component is fabricated from materials grown in the central zone of the SAG region, it will not suffer from thickness variation. The cross-sections of the SAG regions obtained from SAG masks' width of 5 µm and 30 µm were analyzed by TEM (Figure 8c,d). The latter confirm the thickness variation when $W_m$ increases and the excellent material quality of the selectively grown MQW active regions.

### 4.3. Multi Wavelength Laser Array Grown on InPoSi

Based on these MQW-based active regions obtained by SAG, five laser structures were simultaneously grown on InPoSi. The first laser was made in the field area, far from the four other pairs of selective masks, for which $W_m$ values were set to be 5 µm, 15 µm, 20 µm and 30 µm. In the field area, the grown laser structure is composed of a 500 nm thick n-doped InP contact layer, a 6 period AlGaInAs MQW-based active region, a 2 µm-thick p-doped InP cladding layer and a 200 nm-thick p-doped InGaAs contact layer. Then, five shallow-ridge

lasers were fabricated. Finally, 500 μm long bars containing all the lasers were cleaved and soldered on thermally conductive submounts. A photograph and a microscopic image of the laser array are shown respectively in Figure 9a,b. A facet of one of the shallow-ridge lasers in the array was observed by SEM, as shown in Figure 9c.

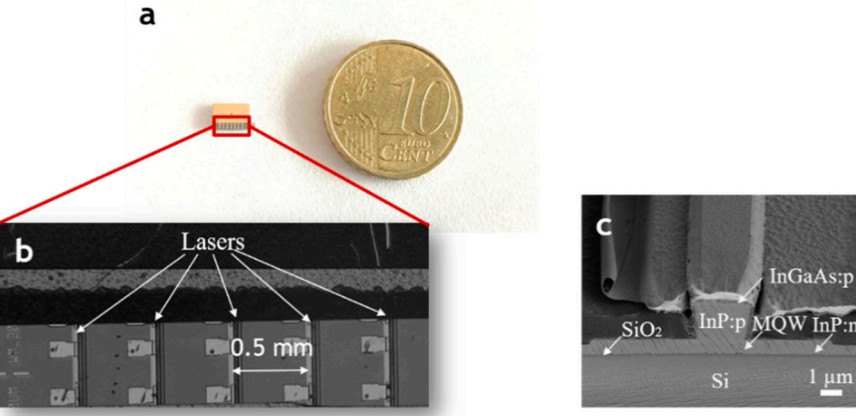

**Figure 9.** Selectively grown laser array on silicon: (**a**) photograph of the chip mounted on a thermally conductive submount; (**b**) Microscopic image of the laser array; (**c**) Cross-section of a shallow-ridge laser observed by SEM.

The emitted signals were coupled by a standard lensed optical fiber to an optical spectrum analyzer Anritsu MS970 and are shown in Figure 10a. Emission spectra are peaking at 1515 nm, 1580 nm, 1610 nm, 1635 nm and 1670 nm for different SAG areas. This covers a 155 nm-wide spectral band including the whole C+L band under continuous-wave operation. The L-I characteristics measured using a broad-area detector in CW operation at 20 °C are presented in Figure 10b. Laser threshold currents remain below 30 mA at 20 °C for the lasers emitting from 1515 nm up to 1610 nm. At least 17 mW of output power is achieved for 200 mA injected current for those lasers. I-V curves are shown on Figure 10c. The mean sheet resistance is of 6.6 Ω ± 1 Ω.

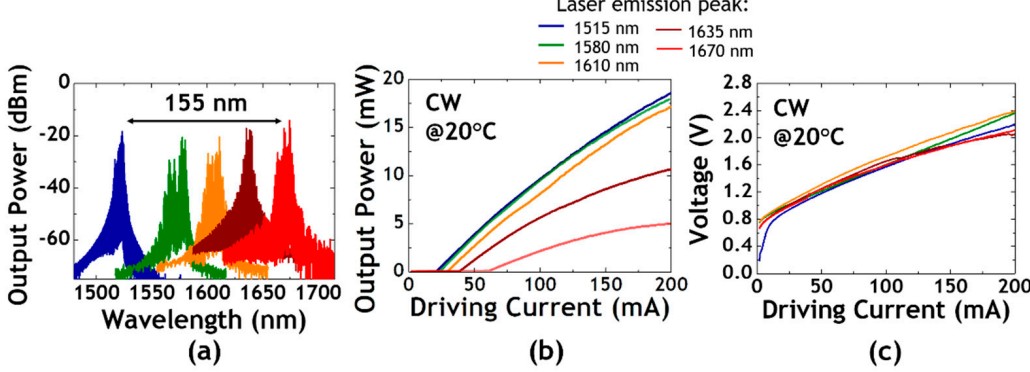

**Figure 10.** Static characteristics measured for the five lasers in the array under CW operation: (**a**) emission spectra measured @20 °C under 100 mA driving current; (**b**) light-current characteristic measured @20 °C; (**c**) current-voltage characteristics measured at 20 °C.

For the longest wavelengths (>1610 nm), the threshold current increased up to 60 mA at 20 °C. This degradation is due to the enlargement of the quantum barrier (QB) and SCH layers in the optical cavity obtained by SAG. This can be also due to the enlargement of the p-doped cladding layer. This undesirable effect could be reduced by lowering the work pressure, hence reducing SAG effect, for the growth of this specific layer. The MQW design could also be optimized for this broad spectral range in order to homogenize laser performance.

Next, the thermal stability of the lasers was tested up to 70 °C under CW operation for the lasers emitting at 1515 nm, 1580 nm and 1610 nm. The results are presented in Figure 11. The lasing effect remains at least up to 70 °C. The laser characteristic temperature $T_0$ is 69 °C for the three lasers. This result demonstrates that these AlGaInAs MQW-based active regions that are selectively grown have similar thermal behavior.

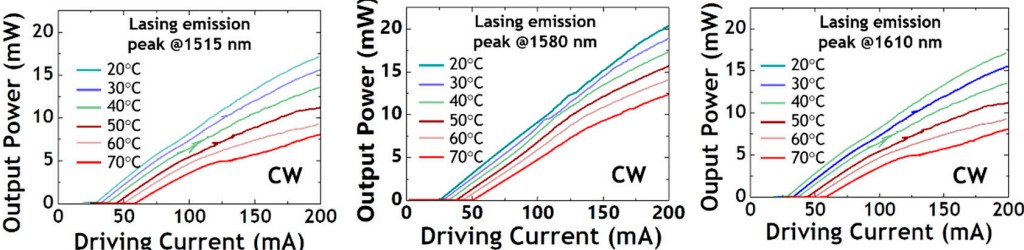

**Figure 11.** LIV Characteristics measured under CW regime up to 70 °C for the lasers emitting at 1515, 1580 and 1610 nm.

## 5. III-V Lasers Grown on InP-SOI

One of the key advantages of the integration platform by InP seed-bonding and regrowth as compared to direct epitaxy on Si is the possibility to use a thin seed layer enabling light-coupling through photonic components underneath the SOI wafer. Therefore, this platform not only allows us to benefit from the multi-regrowth technique applied in the III-V monolithic platform by means of the regrowth technique but also to the Si-photonics circuitry offered by the Si photonics platform [43].

In this section, we show our work on the exploitation of evanescent light-coupling from III-V vertical lasers grown on heterogeneous InP-SOI wafer to Si photonics passive circuitry. To do so, a 2.5 μm-thick III-V laser structure composed of an AlGaInAs-based MQW active region surrounded by two n- and p-doped InP layers was grown on an InP-SOI wafer. After growth, shallow-ridge lasers were fabricated. Two silicon cavity designs were tested: the first design is a FP laser for which cavities are provided by simple cleaved facets and the second design which is III-V laser coupled to DBR cavities made from SiO$_2$/Si Bragg gratings and vertical couplers to extract light.

### 5.1. III-V on SOI FP Laser with Hybrid Facets

The width of the Si waveguide can be adjusted to optimize the optical mode confinement in the MQW. The confinement factor in the Si waveguide as a function of its width is plotted in Figure 12a. The width of the Si waveguide was chosen so that the mode was 80% in the Si waveguide in order to minimize losses induced by the p-doped layers. The optical mode simulation is presented in Figure 12b. A 2.7 mm-long bar was cleaved and soldered on a thermally conductive submount. A cross-section of the hybrid FP laser was observed by SEM in order to control the alignment between the two waveguides (Figure 13a). Then, the laser was characterized by coupling light through a single facet using a broad detector and tested up to 50 °C under continuous-wave regime. The light-current characteristics are shown in Figure 13b. The threshold current measured at 20 °C is of 89 mA with a maximum output power of 2.8 mW. The threshold current is increased up to 160 mA at 40 °C. This result demonstrates the successful mode hybridization between III-V and Si. Laser demonstration is achieved up to 50 °C, which is lower than for the lasers grown on InPoSi. This lower thermal behavior can be attributed to poorer heat dissipation induced by the thickness of the buried oxide (BOX) of 2 μm for InP-SOI as compared to a value of 200 nm for InPoSi. Thermal stability of our components could be further improved by burying the III-V waveguide through InP regrowth steps [26].

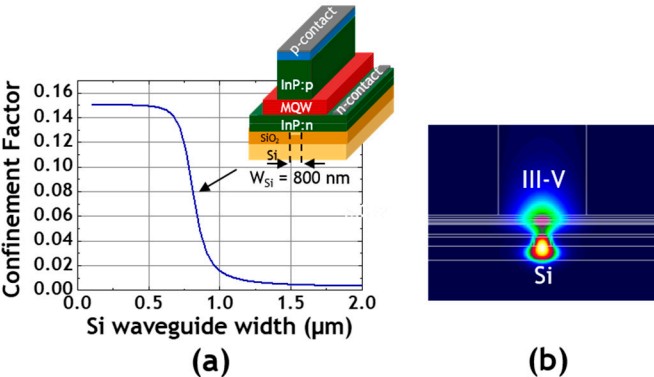

**Figure 12.** (**a**) Evolution of the optical confinement versus silicon waveguide width with Hybrid FP laser description on the top right; (**b**) optical mode simulation obtained for a 800 nm-wide silicon waveguide.

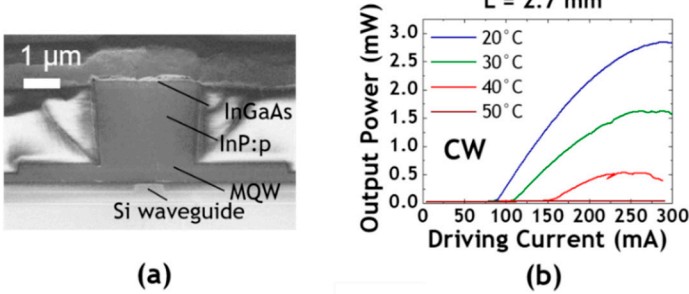

**Figure 13.** (**a**) Cross-section of the FP laser observed by SEM; (**b**) light-current characteristics measured under CW operation up to 50 °C.

### 5.2. III-V on SOI DBR Laser

A III-V/Si laser was then coupled to a Si-photonic DBR cavity in order to form a DBR laser as described in Figure 14. A taper in the Si waveguide was designed to obtain an adiabatic transition between the III-V and the Si waveguides [44]. The light is extracted by means of FCG coupled via a vertical fiber. Bragg mirrors provide cavity reflections with 90% reflectivity for the high reflective mirror and 20% for the low reflective mirror (Figure 15a). Both mirrors were designed to cover a 100 nm-wide band spectrum from 1490 nm to 1590 nm. The light-current characteristics measured at 20 °C under CW regime are presented in Figure 15b. The successful lasing effect demonstrates the efficient light-coupling through Si-photonics DBR cavities. The threshold current is 70 mA with a maximum waveguide-coupled output power of 2.4 mW. The spectra measured at 20 °C for 80 mA injected current with a laser emission wavelength of 1543 nm (Figure 15c). At 120 mA injected current, a red-shift is observed due the self-heating effects. The laser is single-mode for 80 mA injected current and becomes multimode for 120 mA injected current. This behavior is expected considering the wide spectral range covered by the mirrors, which induce lasing of other longitudinal cavity modes for sufficiently high injected current values. Altogether, this proof-of-concept strongly indicates the potential of this integration scheme to benefit from the Si-photonics circuits combined with epitaxial regrowth technique.

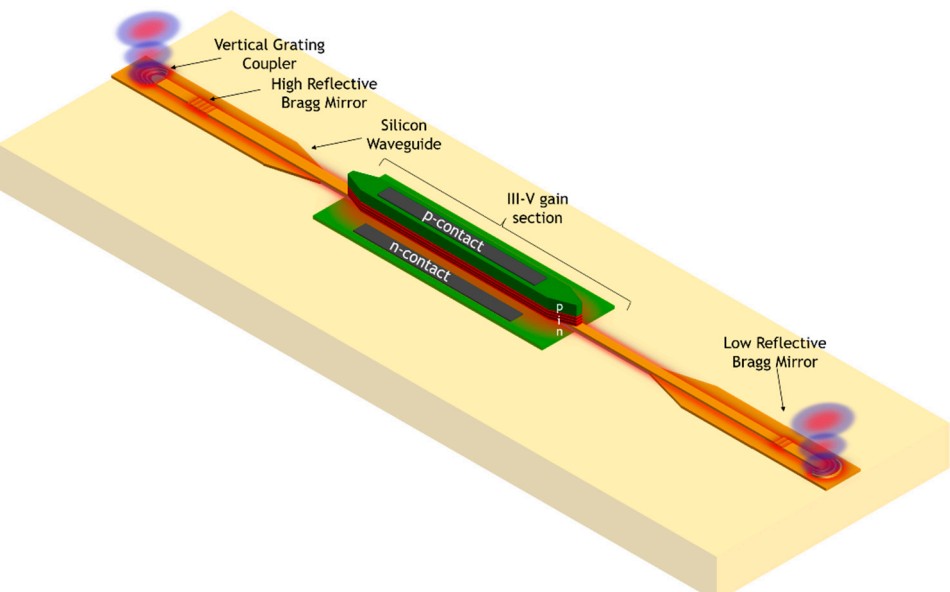

**Figure 14.** 3D Schematic view of the III-V on SOI DBR laser including a III-V gain region coupled to a Si waveguide, Bragg mirrors and vertical couplers.

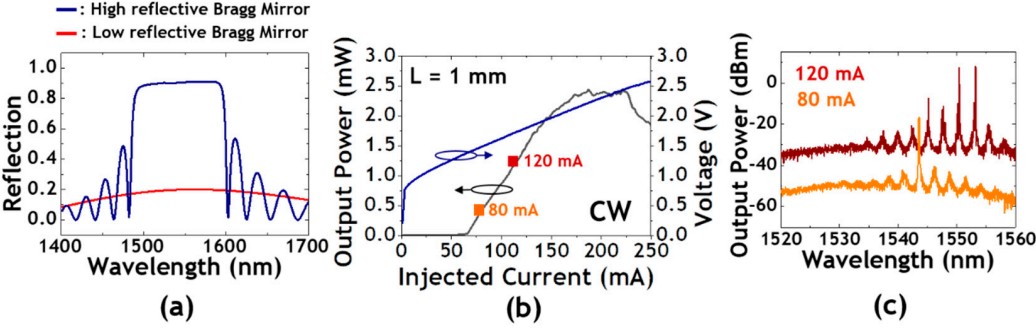

**Figure 15.** (**a**) Simulation of the reflection of the grating mirrors against wavelength with the high reflective mirror in blue and the low reflective mirror in red; (**b**) Waveguide-coupled output power against injected current measured under CW operation @20 °C; (**c**) Spectra measured @20 °C for 80 mA and 120 mA injected currents.

## 6. Conclusions

In this paper, heterogeneously integrated AlGaInAs-based multiple quantum well laser sources on silicon by InP-seed-bonding and regrowth have been presented. The thermal strain assessment from curvature measurement revealed a compressive strain value of 400 ppm at growth temperature. The latter is due to the mismatch of coefficient of thermal expansion between InP and Si. In spite of this strain, a 3 µm-thick laser structure grown on an InP seed-bonded Si substrate (InPoSi) has been demonstrated, providing similar performance to the ones obtained from the same structure grown on a native InP wafer used as a reference. Broad-area lasers obtained from these structures showed a threshold current of 0.4 kA/cm$^2$ under pulse regime at 20 °C and a laser efficiency of 0.092 W/A. Therefore, there is no penalty induced for a 3 µm-thick structure grown on InPoSi in spite of the thermal strain.

In addition, a specific SAG process developed on InP seed-bonded silicon wafers was presented. Selectively regrown multiple quantum well active regions successfully cover a PL range of 160 nm. Using this growth technique, a 5-channel laser array was selectively grown and fabricated. The latter covers a 155 nm-wide spectral range from 1515 nm to 1670 nm. High power operation under continuous wave operation was demonstrated with a maximum of 20 mW for a 200 mA driving current for a 500 µm-long bar. High

thermal stability was shown up to 70 °C for the lasers emitting from 1515 nm to 1610 nm. Improvements could be done to homogenize laser performance over the full spectral band. The design of the active region could be further optimized to cover the broad range achieved by SAG. Therefore, SAG technique has proved its potential for high-density high-performance integration of PICs. Its potential could be further extended for the integration of lasers, modulators and SOAs by tuning their bandgap at will to match the optimum performance of each individual device [39,45].

Last but not least, AlGaInAs MQW-based lasers were successfully grown on heterogeneous InP-SOI wafer. CW lasing operation up to 40 °C has been demonstrated for III-V-on-SOI FP lasers containing hybrid optical modes where 80% of the mode is confined in the Si waveguide. In addition, lasing operation was also demonstrated for a laser grown on InP-SOI coupled to Si-photonic DBR cavities under CW operation at 20 °C. In our case, the use of buried heterostructures such as Buried Ridge Structure (BRS) or even Semi-Insulating Heterostructure (SIBH) could be a path to improve heat dissipation through InP lateral cladding layers [26]. Altogether, this demonstration shows that this integration scheme holds a promising perspective by combining the advanced regrowth technologies developed in the InP platform combined with the benefits of Si photonics platform. Extensive research and development are still needed to explore multiple regrowth on InP-SOI wafers to fully explore the potential of this platform.

Among all III-V/Si integration schemes, the holy grail is to take full benefits of the Si platform in a 200 or 300 mm fabrication line. In this context, applying the regrowth technique on thin InP membrane coupons bonded on large Si wafers using die-bonding approach is appealing [46,47]. In addition, investigating options to reuse the III-V substrate is another key ongoing subject [48]. In this context, the integration scheme by InP seed-bonding and regrowth appears as a promising technique to combine the best of the III-V monolithic integration platform with the benefits offered by the silicon photonics platform.

**Author Contributions:** C.B. contributed to simulation, epitaxy, III-V fabrication, wafer level characterization, methodology, formal analysis, data curation, and writing/original draft preparation. D.N. contributed to III-V fabrication, review and editing. D.M. contributed to III-V fabrication. J.M.R. contributed to simulation, review and editing. G.C. contributed to characterization, review and editing. N.V. contributed to epitaxy, review and editing. D.B. contributed to simulation, review and editing. F.P. contributed to III-V fabrication, review and editing. F.F. contributed to direct-bonding process. C.D. contributed to review and editing. H.M. contributed to sample preparation for TEM analysis. F.B. contributed to review and editing. J.D. contributed to epitaxy, review and editing. All authors have read and agreed to the published version of the manuscript.

**Funding:** This research was funded by H2020 PICTURE project, grant number 780930.

**Institutional Review Board Statement:** Not applicable.

**Informed Consent Statement:** Not applicable.

**Data Availability Statement:** Not applicable.

**Acknowledgments:** The authors would like to thank V. Muffato, S. Malhouitre and L. Sanchez for their technical support and useful discussions. All data supporting this review paper is openly available from the III-V lab repository at R&D Activities (3-5lab.fr).

**Conflicts of Interest:** The authors declare no conflict of interest. The funders had no role in the design of the study; in the collection, analyses, or interpretation of data; in the writing of the manuscript, or in the decision to publish the results.

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
