# Peer review of "AlGaInAs Multi-Quantum Well Lasers on Silicon-on-Insulator Photonic Integrated Circuits Based on InP-Seed-Bonding and Epitaxial Regrowth"

_applsci, doi:10.3390/app12010263_

Round 1
Reviewer 1 Report
The authors provide a thorough and well writen review of their work in hetero-epitaxial growth of InP on silicon wafers using a wafer bonded seed layer. They show that there is no degredation of photoluminesence or performance for lasers grown on the seed layer compared with native substrate. I only have a few questions and minor revisions that I ask the authors to address prior to publication.
1. Please comment on the reason for lower output power as well as higher threshold currents for the hybrid InP/Si lasers. The presence of the BOX may explain the degraded performance at higher temperatures, but what about at room temperature?
2. The AFM image of the InPoSi (Fig 3a) seems to show some patterning or morphology. Is it possible to add a scale bar for height, or otherwise explain the pattern?
3. In this work, the Si or SOI wafer is resized to 3in after InP bonding. Are there any challenges in scaling this up to 200 or 300mm silicon wafer size?
4. The analogous approach to SAG to provide multiple epi structures is multiple-die bonding of InP to Si. For example, see the CWDM-8 transmitter in this paper. Can you comment on any advantages/disadvantages you may see between the two approaches?
Jones, Richard, et al. "Heterogeneously integrated InP\/silicon photonics: fabricating fully functional transceivers." IEEE Nanotechnology Magazine 13.2 (2019): 17-26.
Reviewer 2 Report
The authors provides a review report on the fabrication of AlGaInAs MQW lasers on SOI photonic IC through InP-seed bonding and epitaxial regrowth. This review paper is well summarized and suitable for publication in this multidisciplinary journal. I would suggest the following modifications to improve the depth of the manuscript:
- Please proof-read the whole manuscript one more time. In particular, please take care of the abbreviations to define them at their first use and be consistent for the same term across the manuscript (for instance, silicon or Si).
- Page 3 line 144: Does the "Si top layer" means the Si device layer of SOI wafer?
- Page 3 line 149: One of the main idea of this review paper is the bonding technique, please elaborate and provide more details in the direct bonding process.
- Fig. 1: What is the scale of the acoustic image? Is it taken from the whole Si wafer (200 mm) or just the bonded area (2 in)? It will be helpful if the author mention the scale somewhere in the text or in the inset figure. If it is taken from an area wider than the bonding area, please highlight the bonding area.
- Fig. 2: What is the quantitative magnitude of the growth time? What is the reason of not displaying it? Maybe it is better to display it in the figure to give the readers some knowledge on the process duration to grow such a thick film.
- Fig. 2: Is the InP substrate curved from the beginning? What is the reason and what does it indicate?
- Page 8 line 291: Please elaborate more about the growth temperature difference. This has not been mentioned in any of the previous sections. Can the same device (on InP and Si) quality be produced with the same growth temperature?
- Page 8 line 292: In relation to the previous comment, please also elaborate on how each factors physically influence the threshold current density: cleaved facets and growth temperature.
- Page 9 line 307: What does it mean with "low planarity" as a wafer bonding requirement?
- Page 12 line 423: If possible, please provide some insight on how could possibly these conditions be optimized (which growth condition should be controlled, how to optimize the MQW design, etc).
